# A Convenient, Rapid, Conventional Heating Route to MIDA Boronates

**DOI:** 10.3390/molecules27165052

**Published:** 2022-08-09

**Authors:** Andrew McGown, Anthony K. Edmonds, Daniel Guest, Verity L. Holmes, Chris Dadswell, Ramón González-Méndez, Charles A. I. Goodall, Mark C. Bagley, Barnaby W. Greenland, John Spencer

**Affiliations:** 1Department of Chemistry, School of Life Sciences, University of Sussex, Brighton BN1 9QJ, UK; 2Faculty of Engineering & Science, University of Greenwich, Medway Campus, Central Avenue, Chatham Maritime, Chatham ME4 4TB, UK

**Keywords:** boronic acid, MIDAs, heterocycles

## Abstract

A cheap, conventional, sealed heating reactor proved to be a useful alternative to a microwave reactor in the synthesis of a >20-member MIDA boronate library (MIDA = *N-*methyliminodiacetic acid). Reaction times were 10 min and work-ups were minimal, saving on energy and solvent usage.

## 1. Introduction

Performing chemical reactions in an efficient manner, in terms of reduced solvent and energy use and higher yields, is desirable [1,2]. These include reactions that involve late-stage functionalisation of key scaffolds or ones that “lose control” and produce a greater number of products for greater diversity for biological evaluation [3], e.g., in the synthesis of benzodiazepines or pyridine libraries [4,5,6,7]. Given that time is often a limiting factor, and a significant cost to factor in, processes that are “plug and play” and, can be carried out with little, or no, optimisation, are often de rigueur (Figure 1). 

MIDA boronates occupy a central role in organic synthesis with many applications including, but not limited to, masked boronic acids in total synthesis [8,9,10]; catalysis [11,12,13,14,15,16,17], including iterative or telescopic couplings [17,18,19,20,21]; oxidation chemistry [22] or as C1 or C2 building blocks [23,24]. Although traditionally made by a Dean-Stark protocol, usually employing DMSO as solvent [25,26,27,28], many recent methods have shifted towards milder reaction conditions, more convenient work-up, purification and isolation techniques, notably to enable the synthesis of unstable esters such as 2-phenol [12] or 2-hetero-aryl MIDA analogues [29]. We wish now to disclose a conventional, sealed, heating reactor-based synthesis of MIDA boronates that offers a cheaper, effective alternative to our earlier disclosed microwave-mediated route [30]. Hereafter, we have focused our efforts mainly on a group of “off the shelf” boronic acids that were readily available in our laboratory at the time of the study. Indeed, the samples that were subjected to our new protocol were rather broad in scope, encompassing boronic acids based on an aryl **1**, isoxazole 2, alkyl **3**, benzimidazole, indole, pyrazole **4**–**6**, respectively, or 1,2-methylenedioxybenzene **7** scaffolds (Figure 2).

## 2. Results and Discussion 

For the current program, we made use of a Monowave-50 (Anton Paar), a relatively cheap, albeit low scale, alternative to a microwave reactor, which uses conventional rather than microwave heating (see Experimental Section). Reaction protocols were mainly un-optimised; 10 min, heated to 160 °C, and in DMF, and were subjected to a short work-up (Figure 1). Starting with arylboronic acids with various steric and electronic properties, a small library of aryl MIDA boronate esters was formed in yields ranging from low (30%) to excellent (90%) (e.g., **8a** and **8b,** respectively). This protocol is tolerant of functional groups such as sulphonamide (**8a**, **8n**), ester, nitrile and amide (**8d**, **8f**, **8j** and **8n** respectively). The yield of **8q** is inferior to a recent improved protocol using MIDA anhydride (21% vs. 81%) yet higher than the yield obtained using Dean Stark conditions starting from MIDA (0%) [29]. Similarly, **8r** is formed in inferior yield compared to the recently reported improved protocol (11% vs. 92%, vs. 42% in our previous microwave route) [12,30]. A number of reactions were repeated using PEG-300 as solvent and, in general, gave slightly lower yields, except for **8l**, which was formed in near quantitative yield. All compounds were isolated and fully characterised by ^1^H, ^13^C NMR spectroscopy, and HRMS.

We next focussed on heterocycle-containing boronic acids or an alkylboronic acid and synthesised the analogues **9**–**14** in poor to moderate yields. Analogue **9** was synthesised in lower yield than the state-of-the-art (57% vs. 75%) (Figure 3).

## 3. Materials and Methods

### 3.1. General Conditions

The Anton-Paar Monowave-50 was purchased directly from the manufacturer (https://www.anton-paar.com/uk-en/products/details/synthesis-reactor-monowave-50/, accessed on 1 June 2022). Reactions were performed behind a suitably ventilated, closed, fume hood, in small, bespoke, high-pressure, sealed vials (maximum volume is around 5 mL) on a small scale and needed to be performed by a trained chemist. Solvents, reagents and consumables, such as TLC plates, column material, were purchased from commercial suppliers and solvents/reagents were subsequently used without purification. ^1^H, ^13^C NMR spectroscopy was performed on Varian 500 MHz or 600 MHz spectrometers (Appendix A) and chemical shifts are reported in ppm, usually referenced to TMS as an internal standard. LCMS measurements were performed on a Shimadzu LCMS-2020 equipped with a Gemini® 5 µm C18 110 Å column and percentage purity measurements were run over 30 minutes in water/acetonitrile with 0.1% formic acid (5 min at 5%, 5–95% over 20 min, 5 min at 95%) with the UV detector set at 254 nm. High-Resolution Accurate Mass Spectrometry measurements were taken using a Waters Xevo G2 Q-ToF HRMS (Wilmslow, Cheshire, UK), equipped with an ESI source and MassLynx software. Experimental parameters were: (1)—ESI source: capillary voltage 3.0 kV, sampling cone 35 au, extraction cone 4 au, source temperature 120 °C and desolvation gas 450 °C with a desolvation gas flow of 650 L/h and no cone gas; (2)—MS conditions: MS in resolution mode between 100 and 1500 Da. Additionally, a Waters (Wilmslow, Cheshire, UK) Acquity H-Class UHPLC chromatography pumping system with column oven was used, connected to a Waters Synapt G2 HDMS high-resolution mass spectrometer. 

### 3.2. Experimental Procedures

#### MIDA Synthesis in DMF as Solvent

Typically, a boronic acid (1.0 mmol, 1.0 eq) and *N-*methyliminodiacetic acid (MIDA) (1.0 mmol, 1.0 eq) were dissolved in anhydrous DMF (1 mL) in a Monowave reaction vial containing a stirrer bar. The reaction mixture was heated to 160 °C at full power in the Monowave for 10 min (5 min temperature ramp followed by a 10-min hold time). Upon completion, the reaction mixture was cooled to room temperature and the DMF was removed using an Asynt Smart Evaporator (Isleham, Cambridgeshire, UK), https://www.asynt.com/product/smart-evaporator/, accessed on 1 June 2022. The resulting residue was suspended in water (5 mL) and sonicated for 10 min leading to the formation of a fine precipitate which was collected by filtration. The resulting solid was then suspended in diethyl ether (5 mL) and sonicated for a further 10 min leading to the formation of a colourless solid as pure MIDA protected boronic ester, which was collected and dried by filtration.

### 3.3. Molecules Synthesised

4-(6-Methyl-4,8-dioxo-1,3,6,2-dioxazaborocan-2-yl)benzenesulfonamide (**8a**)



Yield = 93.5 mg (30%). ^1^H NMR (600 MHz, DMSO-*d*_6_) δ 7.78 (d, *J* = 8.1 Hz, 2H), 7.61 (d, *J* = 8.1 Hz, 2H), 7.34 (s, 2H), 4.35 (d, *J* = 17.2 Hz, 2H), 4.14 (d, *J* = 17.2 Hz, 2H), 2.50 (s, 3H). ^13^C NMR (151 MHz, DMSO-*d*_6_) δ 169.7, 144.9, 133.4, 125.1, 62.4, 48.1. HRMS (CI) [M + NH_4_} Predicted mass = 330.0931. Experimental mass = 330.0939. 

2-(3-Fluoro-4-methoxyphenyl)-6-methyl-1,3,6,2-dioxazaborocane-4,8-dione (**8b**)



Yield = 252.7 mg (90%). ^1^H NMR (600 MHz, DMSO-*d*_6_) δ 7.17–7.11 (m, 3H), 4.29 (d, *J* = 17.2 Hz, 2H), 4.08 (d, *J* = 17.2 Hz, 2H), 3.82 (s, 3H), 2.49 (s, 3H). ^13^C NMR (151 MHz, DMSO-*d*_6_) δ 169.8, 152.5 (^1^J_CF_ = 244.0 Hz) 148.1 (d, ^2^J_CF_ = 10.4 Hz), 129.4 (d, ^3^J_CF_ *J* = 3.3 Hz), 119.71 (d, ^2^J_CF_ *J* = 15.3 Hz), 113.7, 62.2, 56.2, 48.0. HRMS [M + H] Predicted mass = 282.0949. Experimental mass = 282.0943. 

2-(3-Ethoxyphenyl)-6-methyl-1,3,6,2-dioxazaborocane-4,8-dione (**8c**)



Yield = 196.9 mg (71%). ^1^H NMR (600 MHz, DMSO-*d*_6_) δ 7.24 (pt, *J* = 7.5 Hz, 1H), 6.95 (d, *J* = 8.0 Hz, 1H), 6.92 (d, *J* = 2.7 Hz, 1H), 6.89 (dd, *J* = 8.0, 2.7 Hz, 1H), 4.30 (d, *J* = 17.2 Hz, 2H), 4.08 (d, *J* = 17.2 Hz, 2H), 3.99 (q, *J* = 7.0 Hz, 2H), 2.50 (s, 3H), 1.30 (t, *J* = 7.0 Hz, 3H). ^13^C NMR (151 MHz, DMSO-*d*_6_) δ 169.9, 158.5, 129.4, 124.8, 118.6, 115.1, 63.1, 62.2, 47.9, 15.2. HRMS (CI) [M + H] Predicted mass = 278.1200. Experimental mass = 278.1200. 

Methyl 4-(6-methyl-4,8-dioxo-1,3,6,2-dioxazaborocan-2-yl)benzoate (**8d**)



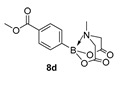



Yield = 221.1 mg (76%). ^1^H NMR (600 MHz, DMSO-*d*_6_) δ 7.92 (d, *J* = 7.8 Hz, 2H), 7.58 (d, *J* = 7.8 Hz, 2H), 4.35 (d, *J* = 17.2 Hz, 2H), 4.13 (d, *J* = 17.2 Hz, 2H), 3.83 (s, 3H), 2.48 (s, 3H). ^13^C NMR (151 MHz, DMSO-*d*_6_) δ 169.8, 166.9, 133.3, 130.4, 128.7, 62.4, 52.6, 48.1. HMRS (ESI) [M + H] Predicted mass = 292.0992. Experimental mass = 292.0995. 

6-Methyl-2-(2-(trifluoromethyl)phenyl)-1,3,6,2-dioxazaborocane-4,8-dione (**8e**)



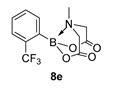



Yield = 114.3 mg (38%). ^1^H NMR (600 MHz, DMSO-*d*_6_) δ 7.74 (d, *J* = 8.0 Hz, 1H), 7.64 (d, *J* = 4.5 Hz, 2H), 7.58 (m, *J* = 8.0, 4.5 Hz, 1H), 4.39 (d, *J* = 17.5 Hz, 2H), 4.17 (d, *J* = 17.5 Hz, 2H), 2.47 (s, 3H). ^13^C NMR (151 MHz, DMSO-*d*_6_) δ 169.7, 136.5, 132.7 (q, ^2^J_CF_ = 30.7 Hz), 131.9, 129.9, 126.4 (q, ^3^J_CF_ = 6.3 Hz), 124.1, 63.8, 49.3. HRMS (ESI) [M + H] Predicted mass = 302.0811. Experimental mass = 302.0811. 

2-Fluoro-4-(6-methyl-4,8-dioxo-1,3,6,2-dioxazaborocan-2-yl)benzonitrile (**8f**)



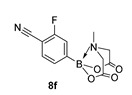



Yield = 198.0 mg (72%). ^1^H NMR (600 MHz, DMSO-*d*_6_) δ 7.89 (pt, *J* = 7.1 Hz, 1H), 7.48–7.44 (m, 2H), 4.37 (d, *J* = 17.2 Hz, 2H), 4.15 (d, *J* = 17.2 Hz, 2H), 2.55 (s, 3H). ^13^C NMR (151 MHz, DMSO-*d*_6_) δ 169.5, 163.3 (^1^J_CF_ = 256.0 Hz), 133.5, 129.8 (d, ^3^J_CF_ = 3.3 Hz), 120.4 (d, ^2^J_CF_ = 17.0 Hz), 114.7, 100.8 (d, ^2^J_CF_ = 15.0 Hz), 62.6, 48.2. HRMS (CI) [M + NH_4_] Predicted mass = 294.1061. Experimental mass = 294.1063. 

2-(3-Ethoxy-4-fluorophenyl)-6-methyl-1,3,6,2-dioxazaborocane-4,8-dione (**8g**)



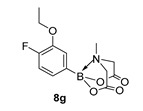



Yield = 208.2 mg (71%). ^1^H NMR (600 MHz, DMSO-*d*_6_) δ 7.19–7.10 (m, 3H), 4.31 (d, *J* = 17.2 Hz, 2H), 4.11 (d, *J* = 17.2 Hz, 2H), 4.09 (d, *J* = 6.9 Hz, 2H), 2.51 (s, 3H), 1.34 (t, *J* = 6.9 Hz, 3H). ^13^C NMR (151 MHz, DMSO-*d*_6_) δ 169.8, 152.6 (^1^J_CF_ = 247.0 Hz), 147.4 (d, ^2^J_CF_ = 10.5 Hz), 129.4 (d, ^3^J_CF_ = 3.3 Hz), 119.8 (d, ^2^J_CF_ = 15.7 Hz), 114.6, 64.5, 62.2, 48.0, 15.1 HRMS (ESI) [M + NH_4_] Predicted Mass = 313.1371. Experimental mass = 313.1376. 

2-(2-Chloro-4-(trifluoromethyl)phenyl)-6-methyl-1,3,6,2-dioxazaborocane-4,8-dione (**8h**)



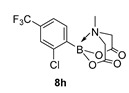



Yield = 245.7 mg (73%). ^1^H NMR (600 MHz, DMSO-*d*_6_) δ 7.80 (d, *J* = 8.0 Hz, 1H), 7.78 (s, 1H), 7.72 (d, *J* = 8.0 Hz, 1H), 4.44 (d, *J* = 17.5 Hz, 2H), 4.19 (d, *J* = 17.5 Hz, 2H), 2.67 (s, 3H). ^13^C NMR (151 MHz, DMSO-*d*_6_) δ 169.6, 138.9, 137.5, 132.1 (q, ^2^J_CF_ = 32.0 Hz), 126.7 (d, ^3^J_CF_ = 3.3 Hz), 123.6 (^3^J_CF_ = 3.3 Hz), 122.9, 64.3, 48.6. HRMS (CI) [M + NH_4_] Predicted mass = 353.0687. Experimental mass = 353.0685. 

2-(2,4-Difluorophenyl)-6-methyl-1,3,6,2-dioxazaborocane-4,8-dione (**8i**)



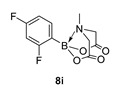



Yield = 209.7 mg, (78%). ^1^H NMR (600 MHz, DMSO-*d*_6_) δ 7.25 (m, 1H), 7.16 (m, 2H), 4.40 (d, *J* = 17.3 Hz, 2H), 4.09 (d, *J* = 17.3 Hz, 2H), 2.63 (s, 3H). ^13^C NMR (151 MHz, DMSO-*d*_6_) δ 169.4, 162.6 (^1^J_CF_ = 237.7 Hz) 159.4 (^1^J_CF_ = 241.1 Hz), 121.0 (d, ^2^J_CF_ = 10.8 Hz), 118.5 (dd, ^2^J_CF_ = 10.2 Hz, ^2^J_CF_ = 10.2 Hz), 117.4 (dd, ^3^J_CF_ = 8.2 Hz, ^3^J_CF_ = 8.2 Hz), 62.9, 48.0. HRMS (ESI) [M + H] Predicted mass = 270.0749. Experimental mass = 270.0743. 

N-(tert-Butyl)-3-(6-methyl-4,8-dioxo-1,3,6,2-dioxazaborocan-2-yl)benzamide (**8j**)



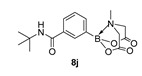



Yield = 129.0 mg (39%). ^1^H NMR (600 MHz, DMSO-*d*_6_) δ 7.79 (s, 1H), 7.75 (dt, *J* = 7.5, 1.5 Hz, 1H), 7.70 (s, 1H), 7.51 (dd, *J* = 7.5, 1.5 Hz, 1H), 7.38 (t, *J* = 7.5 Hz, 1H), 4.33 (d, *J* = 17.3 Hz, 2H), 4.13 (d, *J* = 17.3 Hz, 2H), 2.48 (s, 3H) 1.36 (s, 9H). ^13^C NMR (151 MHz, DMSO-*d*_6_) δ 169.9, 167.3, 135.7, 135.2, 131.7, 128.3, 127.7, 62.3, 51.2, 48.2, 29.1. HRMS (ESI) [M + H] Predicted mass = 333.1622. Experimental mass = 333.1633.

6-Methyl-2-(4-(pyrrolidine-1-carbonyl)phenyl)-1,3,6,2-dioxazaborocane-4,8-dione (**8k**)



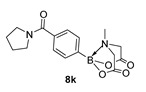



Yield = 83.2 mg (25%). ^1^H NMR (600 MHz, DMSO-*d*_6_) δ 7.46 (s, 4H), 4.33 (d, *J* = 17.2 Hz, 2H), 4.12 (d, *J* = 17.2 Hz, 2H), 3.43 (t, *J* = 7.0 Hz, 2H), 3.34 (t, *J* = 7.0 Hz, 2H), 2.49 (s, 3H), 1.84 (d, *J* = 7.0 Hz, 2H), 1.77 (m, *J* = 7.0 Hz, 2H). ^13^C NMR (151 MHz, DMSO-*d*_6_) δ 169.8, 168.8, 138.1, 132.7, 126.6, 62.3 (2C), 49.3, 48.1, 46.3, 26.4, 24.4. HRMS (ESI) [M + H] Predicted mass = 331.1465. Experimental mass = 331.1472.

2-([1,1′-Biphenyl]-3-yl)-6-methyl-1,3,6,2-dioxazaborocane-4,8-dione (**8l**)



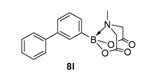



Yield = 281.6 mg (86%). ^1^H NMR (600 MHz, DMSO-*d*_6_) δ 7.66 (m, 3H), 7.63 (d, *J* = 7.5 Hz, 1H), 7.45–7.35 (m, 4H), 7.34 (t, *J* = 7.5 Hz, 1H), 4.33 (d, *J* = 17.2 Hz, 2H), 4.14 (d, *J* = 17.2 Hz, 2H), 2.54 (s, 3H).^13^C NMR (151 MHz, DMSO-*d*_6_) δ 169.9, 141.1, 139.9, 132.0, 131.3, 129.3, 128.7, 127.8, 127.7, 127.3, 62.4, 48.2. HRMS (ESI) [M+H] Predicted mass = 310.1251. Experimental mass = 310.1242. 

2-(3,5-bis(Trifluoromethyl)phenyl)-6-methyl-1,3,6,2-dioxazaborocane-4,8-dione (**8m**)



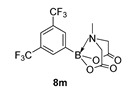



Yield = 311.6 mg (80%). ^1^H NMR (600 MHz, DMSO-*d*_6_) δ 8.10 (s, 3H), 4.40 (d, *J* = 17.2 Hz, 2H), 4.22 (d, *J* = 17.2 Hz, 2H), 2.62 (s, 3H). ^13^C NMR (151 MHz, DMSO-*d*_6_) δ 169.3, 133.3, 129.4 (q, ^2^J_CF_ = 32.6 Hz), 126.4, 124.6, 122.8, 122.7 (q, ^3^J_CF_ = 9.1 Hz), 62.7, 48.1. HRMS (CI) [M + NH_4_] Predicted mass = 387.0951. Experimental mass = 387.0947. 

6-Methyl-2-(4-(pyrrolidin-1-ylsulfonyl)phenyl)-1,3,6,2-dioxazaborocane-4,8-dione (**8n**)



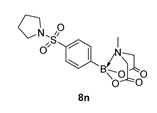



Yield = 326.3 mg (84%). ^1^H NMR (600 MHz, DMSO-*d*_6_) δ 7.78 (d, *J* = 7.8 Hz, 2H), 7.69 (d, *J* = 7.8 Hz, 2H), 4.40 (d, *J* = 17.2 Hz, 2H), 4.18 (d, *J* = 17.2 Hz, 2H), 3.15 (s, 4H), 2.50 (s, 3H), 1.64–1.56 (m, 4H). ^13^C NMR (151 MHz, DMSO-*d*_6_) δ 169.7, 136.9, 133.8, 126.8, 62.4, 48.3, 48.1, 25.1. HRMS (ESI) [M + H] Predicted mass = 367.1135. Experimental mass = 367.1129. 

2-(3-(tert-Butyl)phenyl)-6-methyl-1,3,6,2-dioxazaborocane-4,8-dione (**8o**)



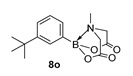



Yield = 230.0 mg (75%). ^1^H NMR (600 MHz, DMSO-*d*_6_) δ 7.44 (s, 1H), 7.37 (dd, *J* = 7.5 Hz, 2.1 Hz, 1H), 7.26 (t, *J* = 7.5 Hz, 1H), 7.19 (d, *J* = 7.5 Hz, 1H), 4.30 (d, *J* = 17.2 Hz, 2H), 4.08 (d, *J* = 17.2 Hz, 2H), 2.46 (s, 3H), 1.26 (s, 9H). ^13^C NMR (151 MHz, DMSO-*d*_6_) δ 169.9, 150.0, 129.9, 129.3, 127.8, 126.1, 62.2, 48.0, 34.8, 31.7. HRMS (ESI) [M + H] Predicted mass = 290.1564. Experimental mass = 290.1568. 

2-(3-(2-Methoxyethoxy)phenyl)-6-methyl-1,3,6,2-dioxazaborocane-4,8-dione (**8p**)



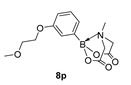



Yield = 199.2 mg (61%). ^1^H NMR (600 MHz, DMSO-*d*_6_) δ 7.25 (pt, *J* = 8.0 Hz, 1H), 6.96 (d, *J* = 7.3 Hz, 1H), 6.94 (d, *J* = 2.7 Hz, 1H), 6.91 (dd, *J* = 8.0 Hz, 1H), 4.30 (d, *J* = 17.2 Hz, 2H), 4.10 (*J* = 17.2 Hz, 2H), 4.08 (t, *J* = 9.0 Hz, 2H), 3.65 (t, *J* = 9.0 Hz, 2H), 3.29 (s, 3H), 2.49 (s, 3H). ^13^C NMR (151 MHz, DMSO-*d_6_*) δ 169.9, 158.4, 129.4, 125.0, 118.7, 115.2, 70.9, 67.0, 62.2, 58.6, 48.0. HRMS (ESI) [M + H] Predicted mass = 308.1305. Experimental mass = 308.1305. 

6-Methyl-2-(perfluorophenyl)-1,3,6,2-dioxazaborocane-4,8-dione (**8q**)



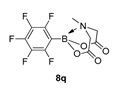



Yield = 68.2 mg (21%). ^1^H NMR (600 MHz, DMSO-*d*_6_) δ 4.22 (d, *J* = 17.2 Hz, 2H), 3.98 (d, *J* = 17.2 Hz, 2H), 2.79 (s, 3H). ^13^C NMR (151 MHz, DMSO-*d*_6_) δ 168.4, 62.6, 45.9. Aromatic carbons not observed [29]. 

2-(3,5-Dimethylisoxazol-4-yl)-6-methyl-1,3,6,2-dioxazaborocane-4,8-dione **9**



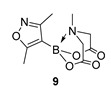



Yield = 154.0 mg (57%). ^1^H NMR (600 MHz, DMSO-*d*_6_) δ 4.32 (d, *J* = 17.2 Hz, 2H), 4.12 (d, *J* = 17.2 Hz, 2H), 2.63 (s, 3H), 2.30 (s, 3H), 2.11 (s, 3H). ^13^C NMR (151 MHz, DMSO-*d*_6_) δ 173.8, 169.5, 162.8, 62.3, 47.5, 12.9, 12.2. [M+H] Predicted mass = 253.0996. Experimental mass = 253.1020. 

6-Methyl-2-propyl-1,3,6,2-dioxazaborocane-4,8-dione **10**



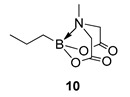



Yield = 34.4 mg (17%). ^1^H NMR (600 MHz, DMSO-d_6_) δ 4.14 (d, *J* = 17.0 Hz, 2H), 3.94 (d, *J* = 17.0 Hz, 2H), 2.80 (s, 3H), 1.31–1.21 (m, 2H), 0.90 (t, *J* = 7.3 Hz, 3H), 0.51–0.45 (m, 2H). ^13^C NMR (151 MHz, DMSO-d6) δ 169.5, 61.9 (2C), 45.9, 17.9, 17.6. [M + H] Predicted mass = 200.1094. Experimental mass = 200.1094. 

6-Methyl-2-(1-methyl-1H-benzo[d]imidazol-5-yl)-1,3,6,2-dioxazaborocane-4,8-dione **11**



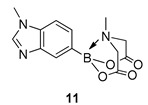



Yield = 161.0 mg (56%). ^1^H NMR (600 MHz, DMSO-d6) δ 8.03 (s, 1H), 7.80 (s, 1H), 7.58 (d, *J* = 8.5 Hz, 1H), 7.43 (dt, *J* = 8.5, 1.2 Hz, 1H), 4.34 (d, *J* = 17.2 Hz, 2H), 4.11 (d, *J* = 17.2 Hz, 2H), 4.02 (s, 3H), 2.45 (s, 3H). ^13^C NMR (151 MHz, DMSO-d6) δ 169.9, 140.6, 133.0, 130.3, 125.8, 123.9, 109.3, 62.2, 48.0, 35.7. [M + H] Predicted mass = 288.1156. Experimental mass = 288.1161. 

2-(1H-Indol-5-yl)-6-methyl-1,3,6,2-dioxazaborocane-4,8-dione **12**



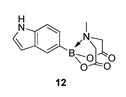



Yield = 142.2 mg (52%). ^1^H NMR (600 MHz, DMSO-d_6_) δ 11.03 (s, 1H), 7.50 (d, *J* = 8.0 Hz, 1H), 7.44 (s, 1H), 7.32 (pt, *J* = 2.8 Hz, 1H), 7.01 (d, *J* = 8.0 Hz, 1H), 6.38 (t, *J* = 2.8 Hz, 1H), 4.30 (d, *J* = 17.2 Hz, 2H), 4.08 (d, *J* = 17.2 Hz, 2H), 2.44 (s, 3H). ^13^C NMR (151 MHz, DMSO-d_6_) δ 170.0, 136.4, 128.6, 125.9, 123.1, 119.9, 115.9, 101.2, 62.0, 47.9. [M + H] Predicted exact mass = 273.1047. Experimental mass = 273.1051. 

6-Methyl-2-(1-methyl-3-(trifluoromethyl)-1H-pyrazol-5-yl)-1,3,6,2-dioxazaborocane-4,8-dione **13**



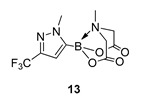



Yield = 21.0 mg (7%). ^1^H NMR (600 MHz, DMSO-d_6_) δ 6.72 (s, 1H), 4.38 (d, *J* = 17.2 Hz, 2H), 4.19 (d, *J* = 17.2 Hz, 2H), 3.93 (s, 3H), 2.66 (s, 3H). ^13^C NMR (151 MHz, DMSO-*d*_6_) δ 169.2, 140.1 (q, ^2^J_CF_ = 37.3 Hz), 124.9 (q, ^1^JCF = 268.3 Hz), 112.2, 62.4 (2C), 47.8. [M + H] Predicted mass = 306.0873. Experimental mass = 306.0875. 

2-(Benzo[d][1,3]dioxol-5-yl)-6-methyl-1,3,6,2-dioxazaborocane-4,8-dione **14**



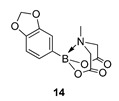



Yield = 199.1 mg (71%). ^1^H NMR (600 MHz, DMSO-*d*_6_) δ 6.90 (s, 1H), 6.89 (d, *J* = 7.9 Hz, 2H), 5.97 (s, 2H), 4.29 (d, *J* = 17.2 Hz, 2H), 4.06 (d, *J* = 17.2 Hz, 2H), 2.49 (s, 3H). ^13^C NMR (151 MHz, DMSO-*d*_6_) δ 169.8, 148.3, 147.4, 126.7, 112.3, 108.6, 100.9, 62.2, 47.9. HRMS (ESI) [M + H] Predicted mass = 278.0836. Experimental mass = 278.0833. 

Boronic acid (1.0 mmol, 1.0 eq) and MIDA (1.0 mmol, 1.0 eq) were dissolved in PEG-300 (1 mL) in a Monowave reaction vial containing a stirrer bar. The reaction mixture was heated to 160 °C at full power in the Monowave for 10 min (5 min temperature ramp followed by a 10 min hold time) (Table 1).

Upon completion, the reaction mixture was cooled to room temperature the resulting reaction mixture was diluted in water (5 mL) and sonicated for 10 minutes leading to the formation of a fine, white precipitate which was collected by filtration. The resulting solid was then suspended in diethyl ether (5 mL) and sonicated for a further 10 min leading to the formation a white solid of pure MIDA protected boronic ester which was collected and dried by filtration.

The following compounds were made by this method.

2-(3-Ethoxyphenyl)-6-methyl-1,3,6,2-dioxazaborocane-4,8-dione **8c**



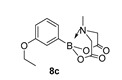



Yield = 152.0 mg (55%). Spectral data as above. 

Methyl 4-(6-methyl-4,8-dioxo-1,3,6,2-dioxazaborocan-2-yl)benzoate **8d**



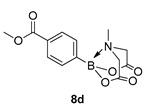



Yield = 193.0 mg (67%). Spectral data as above. 

2-([1,1′-biphenyl]-4-yl)-6-methyl-1,3,6,2-dioxazaborocane-4,8-dione **8l**



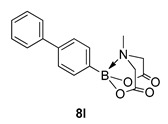



Yield = 309.0 mg (99%). Spectral data as above. 

2-(3-(tert-Butyl)phenyl)-6-methyl-1,3,6,2-dioxazaborocane-4,8-dione **8o**



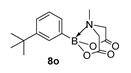



Yield = 299.0 mg (97 %). Spectral data as above. 

6-Methyl-2-(Perfluorophenyl)-1,3,6,2-dioxazaborocane-4,8-dione **8q**



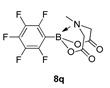



Yield = 68.2 mg (21%). Spectral data as above. 

2-(2-Hydroxyphenyl)-6-methyl-1,3,6,2-dioxazaborocane-4,8-dione **8r**



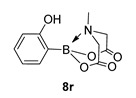



Work-up procedure differed from reported method as, upon the addition of water to the residue, no precipitate formed. The aqueous mixture was extracted into EtOAc (3 × 15 mL). The combined organics were then dried over MgSO_4_ and concentrated to dryness yielding a colourless solid. Yield = 26.2 mg (11%). ^1^H NMR (600 MHz, DMSO-*d*_6_) δ 9.58 (s, 1H), 7.38 (d, *J* = 7.2 Hz, 1H), 7.17 (t, *J* = 7.8 Hz, 1H), 6.81–6.73 (m, 2H), 4.33 (d, *J* = 17.2 Hz, 2H), 4.04 (d, *J* = 17.2 Hz, 2H), 2.63 (s, 3H). ^13^C NMR (151 MHz, DMSO-*d*_6_) δ 169.9, 160.5, 134.5, 130.8, 119.2, 115.0, 63.6, 47.6. [M + H] Predicted mass = 250.0887. Experimental mass = 250.0895.

2-(3,5-Dimethylisoxazol-4-yl)-6-methyl-1,3,6,2-dioxazaborocane-4,8-dione **9**



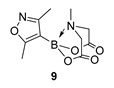



Yield = 80.2 mg (32%). Spectral data as above.

2-(Benzo[d][1,3]dioxol-5-yl)-6-methyl-1,3,6,2-dioxazaborocane-4,8-dione **14**



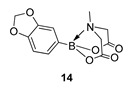



Yield = 187.0 mg (68%). Spectral data as above. 

## 4. Conclusions

We have demonstrated that the use of a relatively cheap conventional heating manifold is capable of generating MIDA boronates in often good to excellent yields. Combined with a short work-up, these protocols should enable facile access to these synthetically useful building blocks.

## Data Availability

Not applicable.

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
