# Peer review of "A Convenient, Rapid, Conventional Heating Route to MIDA Boronates"

_molecules, 2022, doi:10.3390/molecules27165052_

Round 1

Reviewer 1 Report

I think the work is incomplete since the reaction protocols were mainly unoptimized. The main difference from the previous results is that in this case, the authors used Monowave-50 equipment as an alternative to a microwave reactor. In addition, the yields obtained in some cases are lower than those reported in the scientific literature. Although it is of interest in Organic Synthesis, I believe that the article should be rejected and submitted again when the process is optimized.

Author Response

Thank you for the comments.

This shows a cheaper alternative to known routes with some new examples made. Burke repeated his earlier work in Org Lett recently and used a centrifuge as work up. Ours is quicker than our older one, with a simpler work up, is unoptimised but this shows it's useful for making libraries. Yields are, in general, very good.

Reviewer 2 Report

The article A Convenient, Rapid, Conventional Heating Route to MIDA Boronates is continuing of nearly 10 years old paper of the same group. Only microwave heating is substituted by fast conventional heating using heating reactor. If there will be comparism of both method (MW vs. heating reactor) on the same substrates, than it might be informative paper. Unfortunatelly, in this form this article brings basicaly no information. Only that you can do MIDA boronates in DMF with conventional heating. But this is not new. (Muir et al., Organic Letters, 2015, vol. 17, # 24, p. 6030 - 6033; Castro-Godoy et al., European Journal of Organic Chemistry, 2019, vol. 2019, # 19, p. 3035 - 3039; Wang et al., Chinese Journal of Chemistry, 2021, vol. 39, # 7, p. 1825 - 1830; NOVARTIS AG - WO2018/14829, 2018, A1)

Author Response

The reviewer makes some valid points and I thank them. I will add those references, many thanks.

I agree with the microwave comments but this is a new study, using new apparatus e.g. DMF can be conveniently removed using a blowdown apparatus. Actually, we have one overlapping reaction (8r) where this new method is weaker and all the compounds described herein are new (to us); I added that this was made in 42% yield in our Tetrahedron paper from 2012. Cost is significantly less (our microwave broke and we have no new one for comparisons.)

I added those new publication references (ref 15, Castro-Godoy is already cited), Many thanks

Reviewer 3 Report

            The present paper provides an excellent addition to the literature given facile access to valuable reaction intermediates.  Publication subject to minor change is strongly recommended.

            The authors should think about how this paper will read in the fullness of time (or even the shortness of time) once the URL they provide disappears, as certainly will happen in a few months or few years.  No one will have a clue what is meant by an Anton-Paar Monowave-50, or a Monowave reaction vial, or an Asynt Smart Evaporator.  This all reads like corporate “product placement” and is meaningful only for those familiar with those products.  The authors would serve themselves well by providing diagrams and dimensions and specs, even if this needs to be put in the Supporting Information. 

For example, is a Monowave reaction vial thick borosilicate glass or quartz or just a garden variety glass vial? 

What kind of cap is employed to preclude eruption? 

What kinds of safety precautions are taken?

All of these sorts of issues require delineation.  The authors also should think about how this paper will read in the undeveloped world where practitioners will inevitably attempt implementation in an ordinary glass vial in a hot sand bath with no fume hood or protective shields.  Would that be advisable?

The authors absolutely must address safety concerns as outlined above.  Their procedures as written are safe.  But others are not so blessed with this rich equipment and will inevitably carry out bastardized versions with undesired results, and in so doing learn the dangers of “heating liquids in closed volumes.”

Author Response

Thank you. With all due respect, we run NMR on a Varian or Bruker, do mass spec on a Waters or Shimadzu so how can we not name these products as there is sometimes variability? To counter any hints at a conflict of interest, note , that neither company sponsored our work and this was declared. I find these comments very strange. We are simply correctly reporting the protocol and apparatus used.

We have 2 solutions; we name them as per their brand name (and we gave a link e.g. as in the text, to the A.P. Monowave... ) or... we call them "ACME" (as in the classic Coyote vs RoadRunner cartoons!) and leave people wondering how these reactions were performed.  If someone wants to look at these pieces of equipment, they can  search and find them easily. I added a link to the blowdown to help and it is highlighted in yellow in the revised text.

The reaction uses standard glass vials, caps and is sealed and is placed behind a fumehood. it is on a small scale, 5 mL max solvent so should not pose any risk. If people feel like using a sandbath and doing this on a litre scale, they are not following our protocol. it is not to be carried out by untrained chemists, unsafe chemists, children either but I fail to understand the motivation for these questions here? 

I have hopefully been able to address the safety concerns by adding:

Reactions were performed behind a suitably ventilated, closed, fume hood, in small bespoke high pressure sealed vials (maximum volume is around 5 mL) on a small scale and need to be performed by a trained chemist

Reviewer 4 Report

In this manuscript the authors MIDA boronates have been synthesized via a rapid microwave-mediated synthesis of a library of MIDA boronates and reaction time was short and short workup, saving on energy and solvent DMF generating MIDA boronates in often good to excellent yields. The method of choice for the synthesis of synthetically important MIDA boronates such as Aryl, alkyl heterocycle containing small molecules and used in the synthesis of bioactive compounds handle to explore medicinal applications and natural products synthesis.

I suggest accepting this manuscript for publishing in this journal after minor revisions.

Minor edits:

1.      In reference -14 gave wrong address, this is right Eur. J. Org. Chem.2019, 3035–3039

2.       Please refer the manuscript to describe altranative methods of MIDA boronates

(i) J. Am. Chem. Soc. 2007, 129, 6716-6717, DOI: 10.1021/ja0716204

(ii) Org. Biomol. Chem., 2019, 17, 5789, DOI: 10.1039/c9ob00963a.

3.       This manuscript describe was written using poor english make sentences.

Author Response

Thanks for the positive comments. These minor changes have been addressed. However, I feel we have cited sufficient Burke references and his review so we will not add the 2007 citation but we added the OBC one.

Round 2

Reviewer 1 Report

After reviewing the changes made by the authors and carefully reading the Burke paper suggested by the authors (Org. Lett. 2020, 22, 24, 9408–9414), I believe the manuscript can be accepted in its present form.